# Survival of Patient-Specific Unicondylar Knee Replacement

**DOI:** 10.3390/jpm13040665

**Published:** 2023-04-14

**Authors:** Patrick Weber, Melina Beck, Michael Klug, Andreas Klug, Alexander Klug, Claudio Glowalla, Hans Gollwitzer

**Affiliations:** 1ECOM, Arabellastraße 17, 81925 München, Germany; 2ATOS Klinik München, Effnerstraße 38, 81925 München, Germany; 3Dr. Lubos Kliniken München-Bogenhausen, Denninger Straße 44, 81925 München, Germany; 4Knee Centre, Schweinfurter Straße 7, 97080 Würzburg, Germany; 5Praxisklinik Werneck, Balthasar-Neumann-Platz 11-15, 97440 Werneck, Germany; 6König Ludwig Haus, Brettreichstraße 11, 97074 Würzburg, Germany; 7BG Unfallklinik, Friedberger Landstraße 430, 60389 Frankfurt, Germany; 8Klinik und Poliklinik für Orthopädie und Sportorthopädie, Klinikum rechts der Isar, Technische Universität München, Ismaninger Straße 22, 81675 München, Germany

**Keywords:** unicompartmental knee arthroplasty, osteoarthritis, patient-specific implant, partial knee arthroplasty, patient-specific instruments

## Abstract

Unicompartmental knee arthroplasty (UKA) in isolated medial or lateral osteoarthritis leads to good clinical results. However, revision rates are higher in comparison to total knee arthroplasty (TKA). One reason is suboptimal fitting of conventional off-the-shelf prostheses, and major overhang of the tibial component over the bone has been reported in up to 20% of cases. In this retrospective study, a total of 537 patient-specific UKAs (507 medial prostheses and 30 lateral prostheses) that had been implanted in 3 centers over a period of 10 years were analyzed for survival, with a minimal follow-up of 1 year (range 12 to 129 months). Furthermore, fitting of the UKAs was analyzed on postoperative X-rays, and tibial overhang was quantified. A total of 512 prostheses were available for follow-up (95.3%). Overall survival rate (medial and lateral) of the prostheses after 5 years was 96%. The 30 lateral UKAs showed a survival rate of 100% at 5 years. The tibial overhang of the prosthesis was smaller than 1 mm in 99% of cases. In comparison to the reported results in the literature, our data suggest that the patient-specific implant design used in this study is associated with an excellent midterm survival rate, particularly in the lateral knee compartment, and confirms excellent fitting.

## 1. Introduction

Unicompartmental knee arthroplasty (UKA) in isolated medial or lateral osteoarthritis leads to good clinical results. In comparison to total knee arthroplasty (TKA), surgery can be performed through a shorter approach, leading to quicker rehabilitation, and the kinematics after implantation of UKA are similar to those of the physiological knee [1,2,3,4]. Good clinical results were confirmed in two recent randomized controlled trials [5,6]. Registry Data confirmed these results and showed that the Oxford knee score is higher in patients with UKA compared to TKA [7]. On the other hand, the revision rate in UKA is nearly twice as high as for TKA. In the German arthroplasty registry, for example, the revision rate for UKA was 8% after 7 years compared to 4% in TKA [8].

There are many reasons for revision of UKA. The optimal positioning of UKA has been studied extensively [9,10,11]. In this respect, free-hand implantation of UKA leads to up to 41% of outliers of the optimal range [12]. Other reasons for revision are complications associated with tibial overhang or undersizing. Tibial undersizing may increase the risk of implant migration into the softer cancellous bone with consecutive loosening. On the other hand, a recent analysis showed that a tibial overhang over the bone of more than 3 mm can lead to a revision rate of up to 20% [13]. Medial overhang of the prosthesis is sometimes difficult to avoid. The placement of the medial unicondylar prosthesis is limited in the lateral direction, as harm to the anterior cruciate ligament has to be avoided. Choosing a smaller implant can lead to undercoverage in the antero-posterior direction.

Patient-specific implants (PSI) are produced individually for every patient based on a computed tomography scan of the leg. They have shown a better coverage of the tibia in CAD studies, with 0% overhang in comparison to off-the-shelf implants, which show overhang of up to 70% [14]. Furthermore, it has been shown that the implantation of the PSI in combination with patient-specific instruments leads to reproducible and precise implantation [15]. Thus, PSI should help to avoid suboptimal implantations leading to failures of UKA [12,16].

Lateral UKA can lead to good clinical results in isolated lateral osteoarthritis of the knee [17]. The procedure is performed less frequently and the revision rate is reported to be much higher than in medial UKA, with a revision rate of 12% after 5 years [18]. A reason for this is that lateral UKA is technically more challenging than medial UKA due to the lower number of indications, as well as the different functional anatomy of the lateral compartment. One more reason is the fact that most of the available UKA systems offer no specific lateral implants. Instead, the medial tibial component of one side (left/right) is used as a lateral component on the contralateral side. Knowing that the biomechanics of the lateral component differs to that of the medial, this is probably one reason for the higher revision rate of lateral UKAs [17]. With patient-specific implants a better fitting for lateral prosthesis as well is awaited.

The use of a patient-specific unicompartmental knee prosthesis should result in more precise implantation and better coverage. These advantages should lead to a lower revision rate. However, clinical data showing this are sparse. The aim of this retrospective study was to analyze the survival of more than 500 PSI UKA and to measure the overhang of the tibial component.

## 2. Materials and Methods

A total of 537 consecutive knees in 492 patients that received isolated medial or lateral patient individual UKA (iUni, ConforMIS, Billerica, MA, USA), were included in the study. Surgeries were performed between 09/2010 and 03/2020 in three centers (ECOM Munich, Germany, Knee Centre Würzburg, Germany, and Klinikum rechts der Isar der Technischen Universität München, Germany) by three different surgeons (MK, HG, PW). There were 507 medial prostheses (462 patients) and 30 lateral prostheses (30 patients). Inclusion criteria were patients with anteromedial or lateral osteoarthritis of the knee or avascular osteonecrosis of the medial femoral condyle (AVON, Morbus Ahlbäck) as well as knee pain exclusively localized to the affected compartment.

Exclusion criteria were the following:Lateral or medial chondromalacia Grade III or more, or symptomatic retropatellar osteoarthritisROM < Flexion/Extension 100–10–0°Varus/Valgus deformity (hip–knee–ankle angle) > 15°Patients with valgus knees and medial osteoarthritis or patients with a varus knee in lateral osteoarthritisStatus after osteotomyLigament insufficiencyAllergy against metal ions (Ni, Co, Cr)

### 2.1. Prosthesis and Surgical Technique

In all cases, the Conformis iUni knee was implanted. Every patient had a preoperative computed tomography scan of the knee and of the hip and ankle. Planning was performed individually for every patient according to the individual anatomy. The implant was delivered to the surgeon in combination with an iView surgical plan (Figure 1).

In brief, the prosthesis was implanted for the medial knee through a limited medial parapatellar approach, and for the lateral compartment through a lateral parapatellar arthrotomy. After exposition of the joint and removal of the meniscus, the joint is exposed and isolated medial or lateral osteoarthritis is confirmed. The functional integrity of the anterior cruciate ligament is checked. After this, the rest of the chondral layer on the medial or lateral femoral condyle as well as the osteophytes as indicated on the iView Surgical plan are removed. This step is crucial for correct placement of the individually designed instruments, since the surgical plan is based on the CT scan and therefore on the bony surfaces only. Correct position of the femoral jig (patient-specific instrument) is confirmed by comparison with the surgical plan. The next step consists of removing both the complete remains of the tibial cartilage and the marked osteophytes. Four different heights of balancer chips (1 mm steps) can be inserted into the knee to achieve an appropriate ligament tension. The ligament tension must be appropriate in extension. On the medial side, a laxity of 1–2 mm is aimed on the lateral side of 2–3 mm. After achieving correct ligament tension, the tibial cutting guide is put on the selected balancer chip seating on the tibia. The correct position of the cutting guide is additionally confirmed by an alignment rod attached to the tibia that has to be parallel to the tibial crest. The tibial resection can be performed after fixation of the tibial cutting guide. After removal of the tibial bone, the 8 mm spacer (height of the tibial component and the inlay) is positioned into the knee and the femoral jig is positioned on the femoral bone and in contact with the spacer block. With this technique, the position is achieved in accordance with the bone and the ligament tension. After fixation of the femoral jig, the dorsal femoral resection can be performed. There is no distal femoral resection as the implant is designed to replace only the distal femoral cartilage. Next, the trial is introduced and the joint play is evaluated over the complete range of motion. If satisfactory, the tibial preparation is finished, and the bone is prepared for cementation. Original implants are always cemented with a fixed bearing inlay. If there is excessive joint laxity, a 2-millimeter-higher inlay is available [19].

### 2.2. Patient Follow-Up and Data Collection

All patients are regularly followed-up clinically and radiologically after joint arthroplasty in the three centers (after 6 weeks, 1 year, and then every 2 years). At every control visit, a clinical examination as well as radiography of the knee in two planes are performed. If patients do not show up to the appointment, they are reminded by phone call. If they cannot come to the appointment, they are asked by phone if the prosthesis is still in situ or if any revision surgery was performed. If patients do not answer, a letter is sent asking them to contact the physicians’ office. Revision surgery was defined as exchange arthroplasty of the inlay or the femoral and/or the tibial implant components.

For the study purposes, an evaluation of the patient’s charts and already collected data was performed. After all the data were documented for each patient, an irreversible anonymization was undertaken. Ethical approval was obtained prior to the study (Ethikkommission an der Technischen Universität München, Germany, Study 250/21 S-EB). As only a retrospective analysis of already collected data was undertaken with irreversible anonymization, informed consent of the patient was waived by the local ethics committee.

In the study, a minimal follow-up of one year was required. The survival of the prosthesis was assessed, and Kaplan–Meier curves were calculated. A sub-analysis of medial and lateral UKAs was also performed.

Furthermore, antero-posterior respective medial and lateral overhang of the tibial component of the prostheses were measured on the immediate postoperative X-rays.

### 2.3. Statistical Analysis

All statistical analyses were performed using SPSS version 25 (SPSS, Armonk, NY, USA). Descriptive analyses are reported as means, SDs, and ranges for continuous variables, and frequencies and percentages for discrete variables. Overall survivorship was determined using the Kaplan–Meier method.

## 3. Results

### 3.1. Preoperative Data

The preoperative demographic variables of the patients, such as the radiographic state of the osteoarthritis according to the Kellgren and Lawrence classification [20], are shown in Table 1.

### 3.2. Follow-Up

In total, 512 prostheses were available for follow-up (95.3%) at a mean of 4.5 years after surgery (1–10.8 years). Two patients had died (0.2%) and twenty-three (4.5%) were not available for follow-up due to different reasons, such as having disconnected telephone numbers or not answering on multiple attempts. In the patients with medial UKA, the follow-up rate was 95.7% (485/507 patients) at a mean time of 4.6 years (SD 2.4) after surgery. In patients with lateral UKA, the follow-up rate was 90% (27/30 patients) at a mean of 4.2 (SD 2.5) years.

### 3.3. Survival of the Prosthesis

#### 3.3.1. Overall Survival

Survival of the iUni UKA (both lateral and medial) is shown in Figure 2 Overall, survivorship after 4.5 years without revision for any reason was 96.0%.

The reasons for revision are given in Table 2.

If only revisions for mechanical failure (aseptic loosening, wear, and periprosthetic fracture) are considered, the survival rate after 4.5 years was 97.5% (Figure 3).

#### 3.3.2. Survival of the Medial UKA

Of the medial UKAs, 20 revisions out of 485 patients were performed after a mean of 4.5 years, corresponding to a survival rate of 95.8% (Figure 4).

#### 3.3.3. Survival of the Lateral UKA

The 4.2-year survivorship for the 27 lateral UKAs was 100%. There was no revision of any lateral UKA.

### 3.4. Reasons for Revision

In total, 20 revisions were performed. In nine cases, there was an aseptic loosening leading to revision, and in five cases an infection was the reason for revision. The reasons for revision are displayed in Table 2.
jpm-13-00665-t002_Table 2Table 2Reasons for revisions of the iUni arthroplasty (all medial, no revision of lateral knees).Reason for Revision
Aseptic loosening (six isolated tibial, three combined femoral + tibial)9 (1.73%)Infection5 (0.97%)Older periprosthetic fracture1 (0.2%)Tibial bone marrow edema1 (0.2%)Progressive osteoarthritis in the other compartments1 (0.2%)Infrapatellar contracture syndrome (revision at external institution)1 (0.2%)Not reported (revision at external institution)1 (0.2%)


### 3.5. Radiological Analysis

Immediate postoperative X-rays were stored in the patient charts and consequently available for work-up in 431 (80.3%) of the 537 initial patients. In four (0.9%) prostheses, there was a medial tibial overhang of up to 3 mm. None had a relevant anteroposterior overhang. Two prostheses (0.5%) had an overhang of 1 mm, one of 2 mm (0.2%), and one of 3 mm (0.2%).

In 404 (79.7%) patients of the medial group, postoperative X-rays were available, with three (0.7%) prostheses showing an overhang of up to 3 mm.

In the lateral group, X-rays were available in all patients. Of these, there was one patient with a lateral overhang of 2 mm (3%).

Figure 5 shows the postoperative X-ray of a lateral UKA:

## 4. Discussion

The present study evaluated the outcomes of patient-specific UKA for isolated medial or lateral osteoarthritis. Although UKA yields good clinical outcomes, revision rates are relatively high compared to total knee arthroplasty, partly due to poor fitting of conventional off-the-shelf prostheses, resulting in possible overhang of the tibial component over the bone in up to 20% of cases.

This retrospective study analyzed 537 patient-specific UKAs (507 medial and 30 lateral) implanted in three centers over a decade, with a minimal follow-up of 12 months (range: 12–129 months), and is the largest available study on patient-specific UKA. In essence, this study showed a high survival rate in patient-specific unicondylar knee replacement of 96% in 512 knees and of 97% if considering mechanical failure alone at a midterm survival of 4.5 years. Moreover, the theoretical advantage of an excellent fitting of the tibial component of prosthesis to the bone [14] was also shown, with less than 1% of patients showing a tibial overhang of more than 1 mm.

The UKA revision rate is higher compared to TKA. In the most recent report of the German Arthroplasty registry, a revision rate of 7% is reported for UKA after 5 years [8]. In the Australian registry (AAONR), the revision rate at 5 years is comparable with 6.5% and also double the TKA revision rate, which is also the case in the NJR [21,22]. In comparison to these registry data, the present study showed favorable results for an individually designed UKA, with a revision rate of 4% at 5 years. Furthermore, the most impactful data investigating implant survival are currently retrieved from joint replacement registries, since very large numbers can be assessed over time. However, thus far, no registry data have been available for patient-specific UKAs. This emphasizes the importance of performing individual studies with large patient numbers and high follow-up rates. The present study is the largest analysis of the iUni implant, with more than 500 cases involved and a follow-up rate of 95.3%. Thus, the present study is—although limited by its retrospective character—the most robust analysis currently available of implant survival of the patient-specific UKA.

The PSI technique can be compared to modern robotically assisted implantations. A recent study of one center with 1000 knees showed a very high survival rate at 5 years for robotically assisted UKA of 98% excluding inlay exchanges [23]. This survival rate is approximately comparable to the 97% survival rate considering mechanical failure alone observed in this study.

The good survival of the robotically assisted UKA is confirmed in a recent study with data of the Australian registry (AAONR). At 3 years, the robotically assisted UKA had a revision rate of 2.6%, which was half that of the non-robotic UKA (5.0% at 3 years). The best-performing non-robotic UKA reached a revision rate of 3.7% [24]. Again, the PSI of this study showed results comparable to the robotically assisted UKAs and the best non-robotic UKA implant.

There are, to the knowledge of the authors, two studies reporting the results of patient-specific UKA. In the study of Pumilia, 349 knees (same implant as in the present study) were analyzed at a follow-up of 4.8 years with a survival rate of 97.8%, which was slightly better than the results of this study. However, the follow-up rate was less than 70%, which is a potential bias and could have influenced the results [25]. A smaller study also using the iUni by Conformis reported a 100% survival rate of 31 medial UKA after a short-term follow-up of 2.4 years [26].

The present study also included 30 lateral UKAs in 30 patients with a survival rate of 100% at 4.2 years. The lateral compartment of the knee is biomechanically and anatomically different from the medial compartment. Most commercially available unicompartmental implants are not designed specifically for the lateral compartment and therefore the fitting of the prosthesis in the lateral compartment is even more difficult. Furthermore, lateral UKA is performed less frequently, which makes it also more challenging. The literature with follow-up of more than 5 years is sparse, reporting a survival rate of 84–100% for fixed-bearing knees and 79–92% for mobile-bearing knees [27]. The analysis of registry data from the National Joint Registry for England, Wales, Northern Ireland and the Isle of Man revealed 93% survival of 2052 lateral UKAs at 5 years [28]. In contrast, a study by Demange et al., using the same lateral PSI implant also used in the present study, in 33 patients showed a high survival of 97% at 3 years and a better tibial fitting in comparison to a conventional implant. The survival in the conventional group of lateral UKA in the mentioned study was 85% [29]. The present study confirms the favorable results of the PSI, especially in lateral unicompartmental osteoarthritis, in a limited number of patients.

Tibial fitting of the prosthesis is important, as Chau et al. found that an overhang of >3 mm resulted in poorer clinical outcomes on the medial side [30]. In their study, they found an overhang >3 mm in 10% of the patients. Undersizing is also not desirable as the prosthesis will be placed only on weaker cancellous bone, increasing the risk of implant migration and loosening. In a more recent analysis, it was even shown that an overhang of more than 3 mm leads to an increased revision rate of up to 20%, compared to 3% in patients with minor overhang at a follow-up time of five years [13]. The present study showed a very good fitting of patient-specific prostheses, with no overhang of more than 3 mm and only 1% with more than 1 mm. Thus, our study may corroborate the hypothesis that avoidance of tibial overhang is correlated to higher survival rates. The good fitting of the patient-specific tibial component should thus improve survival rate and clinical outcomes.

One possible concern in PSI is the radiation to which patients are exposed through the preoperative computed tomography (CT) scan. Modern CTs have an effective dose between 1 and 7 millisievert (mSv), depending on the organ and the technique. The effective dose for a CT scan of the knee is reported to be 1.3 mSv [31]. In the protocol for PSI, a few slides have to be conducted on the hip joint with a slightly higher dose. The average effective dose through environmental sources is estimated to be 2.4 mSv per year in central Europe, ranging from 1 to 10 mSv depending on activity and the exact living area. Radiation through medical exposures is on average an extra 2 mSv, with variations depending on age and medical condition. The applied radiation of the preoperative CT scan is not negligible. However, it has to be weighed against the potential advantages of PSI. If the implants lead to lower revision rates, there will be reduced radiation for patients that do not need multiple X-rays before and after revision surgery. Furthermore, in conventional knee arthroplasty, preop whole-leg X-ray is mandatory. The radiation of these images is not negligible either. This radiation is not necessary in patients receiving PSI implants, since the leg axis is also determined through CT. Finally, in the opinion of most experts, the risk of developing a disease through CT scanning of the thorax or the abdomen in patients aged over 65 years is negligible [32,33]. Therefore, the much-lower radiation dose of a CT scan of the extremity is probably irrelevant in these patients. Considering the potential advantages of PSI, the necessary radiation for a preoperative CT scan to plan the PSI implants is justifiable in the eyes of the authors.

In spite of the large patient sample, this study has some limitations. First, it was a retrospective analysis with no comparison group. However, it was a consecutive series with a large number of included patients. The follow-up rate of more than 95% is also very high, resulting in a robust data set.

Second, the number of patients receiving a lateral UKA was relatively small. This is due to the significantly rarer indication of lateral UKA. Large case numbers can most likely be obtained with registry analysis, which should also become available for patient-specific implants in the future.

A further limitation of the use of PSI is the costs that are 2–2.5-fold higher than for conventional implants, depending on the country. On the other hand, in PSI there is no need for additional trays, which reduces costs of sterilization and logistics and saves time. If PSI will lead to reduced revisions, as is the case in the present study, there is another potential of saving money by investing more in the implant during primary surgery. In the future, the costs of the implants should also be less by reducing the costs for production through modern 3D printing and an eventually higher use of PSI, which should also reduce costs. Considering all these facts, the costs are probably only slightly higher than in conventional implants, although detailed information of the exact extra costs is missing.

Furthermore, the observed mean follow-up is only 5 years. Nevertheless, it is important to analyze the results at this point also, as eventual advantages or disadvantages of a device and potential risks can already be observed earlier. The present study showed that the survival rate was better in comparison to most of the used UKA at mid-term follow-up, and it is also likely that this difference will be observed in the longer term, justifying continuous use of patient-specific UKA.

Finally, the study does not allow conclusions about the functional results, since no clinical scores were included in the analysis. This has not been the aim of the study, since implant survival should be investigated. There have been many studies showing excellent clinical results in unicompartmental knee replacement [25,26,34], including the implant used in this study. The patients of this study are followed very closely in the centers after UKA, and the low revision rates suggest that the clinical results are also satisfactory. Nevertheless, future studies including clinical results are mandatory.

## 5. Conclusions

The present study is the largest analysis of patient-specific UKA, with more than 500 prostheses analyzed retrospectively. The survival rate of 96% at 4.5 years (97.5% if considering mechanical failure alone) is excellent in comparison to the literature, and comparable to robotic-assisted UKA. Lateral UKA is a more complex procedure with higher risk of revision. The use of a patient-specific implant in this study showed a 100% survival rate at 4.2 years in 30 lateral knees, and these results should be confirmed in the future on a higher number of patients.

## Figures and Tables

**Figure 1 jpm-13-00665-f001:**
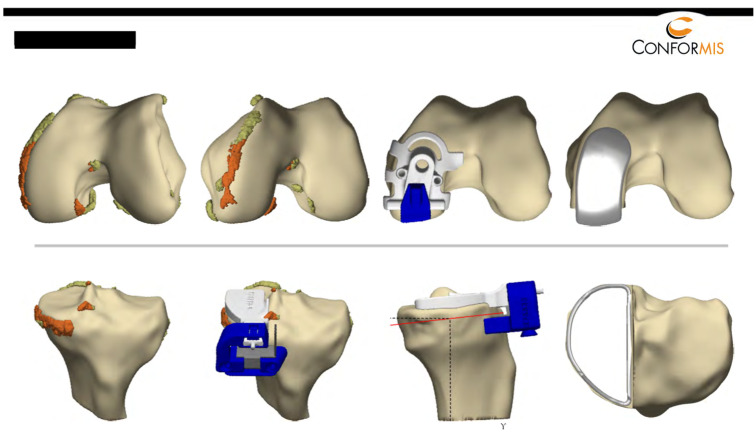
Preoperative planning of the prosthesis and surgical guide (Iview) delivered for every patient. It shows the osteophytes that must be removed to position the patient-specific instruments (orange-colored). The position of the patient-specific instruments is also shown. In particular, the position of the femoral jig in accordance with the femoral component is very helpful for the surgeon. Furthermore, it shows the final position of the prosthesis (see further details in the text).

**Figure 2 jpm-13-00665-f002:**
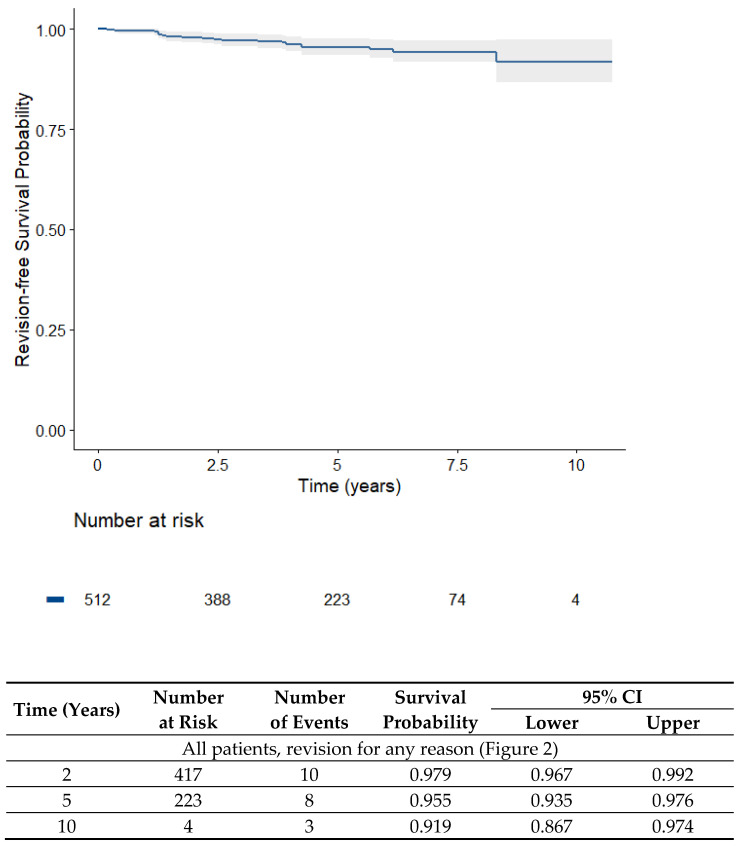
Kaplan–Meier survival analysis of all patients (512 lateral and medial unicondylar knee prostheses) over time (overall survival with revisions for any reason).

**Figure 3 jpm-13-00665-f003:**
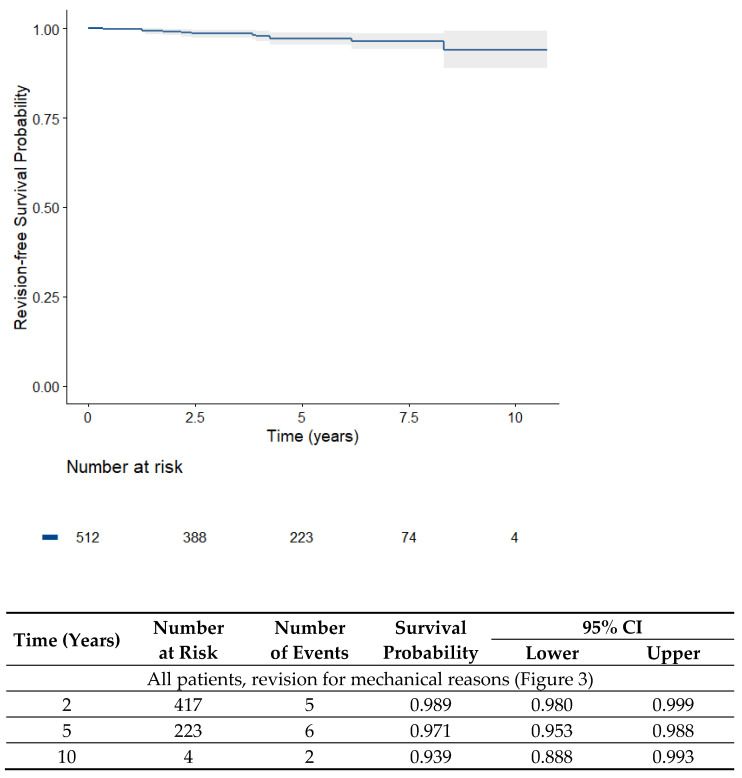
Kaplan–Meier survival analysis of all UKA (512 medial and lateral unicondylar knee prostheses) over time (overall survival with revisions for mechanical reason).

**Figure 4 jpm-13-00665-f004:**
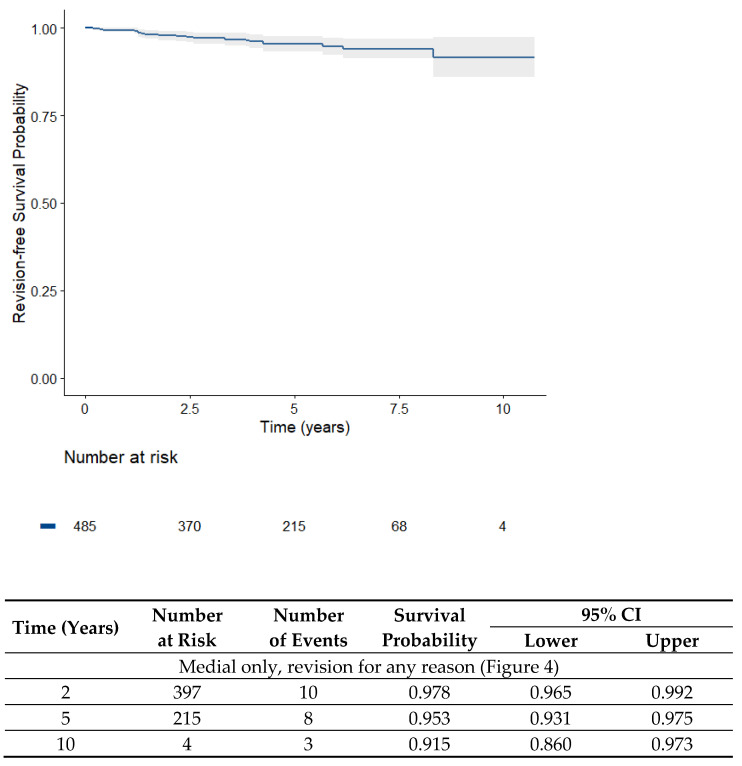
Kaplan-Meier survival analysis of the medial UKAs (485 medial unicondylar knee prosthesis) over time (overall survival with revisions for any reason).

**Figure 5 jpm-13-00665-f005:**
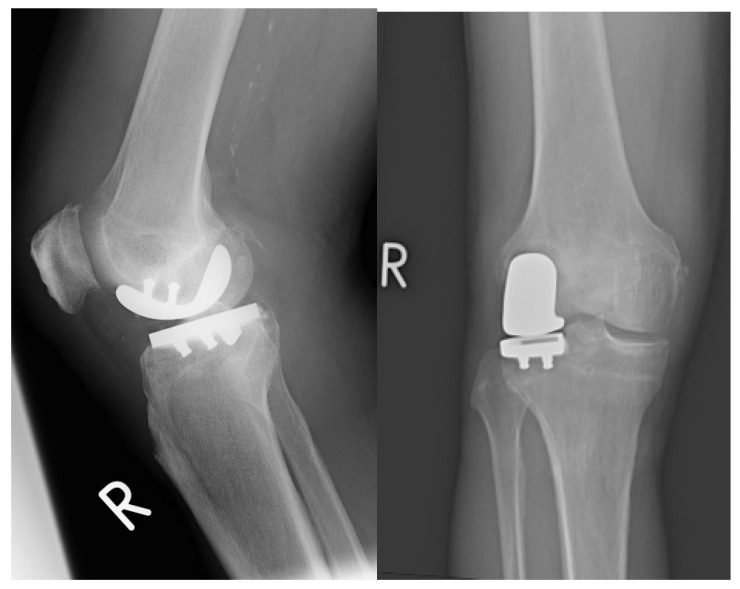
Postoperative X-ray of a patient-specific lateral unicondylar knee prosthesis.

**Table 1 jpm-13-00665-t001:** Preoperative characteristics of the patients.

Variable	Total (*n* = 537)	Medial UKA (*n* = 507)	Lateral (*n* = 30)
Mean age, years (SD)	66.6 (9.4)	66.9 (9.9)	60.9 (9.4)
Male sex, *n* (%) Mean BMI, kg/m^2^ (SD)	313 (58.3) 29.2 (4.9)	299 (59.0) 29.4 (4.9) ^1^	16 (53.3) 26.2 (3.6)
Preoperative KL ^1^ grade, *n* (%)			
Grade 1 to 2	0.2% *	0.2%	0
Grade 3 to 4	99.8	99.8	100

^1^ Kellgren and Lawrence. * Patients with avascular necrosis of the femoral condyle.

## Data Availability

Since this was an observational study, no study registration was undertaken.

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
