# Peer review of "Survival of Patient-Specific Unicondylar Knee Replacement"

_jpm, 2023, doi:10.3390/jpm13040665_

Round 1

Reviewer 1 Report

The article presents a five-year follow-up of PSI in UKA of a group of 500 prostheses. Only mechanical loosening is considered. Fixation stability after 4.5 years was reported at 97.5%, which is comparable to robotic assisted UKA.
The topic is presented convincingly, based on the literature cited by the authors. The topic is comprehensively examined and the results are presented correctly. Applicable outcome assessment scales and popular statistical methods are used. The statistical analysis is precise and wide-ranging.
The manuscript is clearly written and well structured. Recent publications related to the problem are cited. Figures and tables illustrate well and support what is presented in the text.
The discussion is correct and comprehensive. The conclusion is logical and useful for the reader.

Author Response

  Dear Editor, dear reviewers,

 We thank you for your useful comments, which clearly improved and clarified the manuscript. We have thoroughly revised the manuscript and hope we could address all of the aspects you have raised. The new text of the revised manuscript is in “track changes”. Furthermore, the detailed comments and corrections are listed below point by point:

Reviewer 1:

The article presents a five-year follow-up of PSI in UKA of a group of 500 prostheses. Only mechanical loosening is considered. Fixation stability after 4.5 years was reported at 97.5%, which is comparable to robotic assisted UKA.
The topic is presented convincingly, based on the literature cited by the authors. The topic is comprehensively examined and the results are presented correctly. Applicable outcome assessment scales and popular statistical methods are used. The statistical analysis is precise and wide-ranging.
The manuscript is clearly written and well structured. Recent publications related to the problem are cited. Figures and tables illustrate well and support what is presented in the text.
The discussion is correct and comprehensive. The conclusion is logical and useful for the reader.

 We thank the reviewer for his comments.

Spell check has been done

Reviewer 2 Report

This is essentially a radiographic retrospective review of a large number of medial compartment UKRs and a small number of lateral compartment UKRs using custom made implants. 

It does show less tibial overhang than when using standard implants, which one would expect. It also shows good mid term implant survival using revision as an end point. 

There are a number of issues which diminish the value of this paper.

Firstly the use of CT preoperatively exposes the patient to increased radiation and I have ethical reservations in this regard.

Secondly , the increased cost must be a consideration in using custom made implants for all cases. I think that any potential benefit would not justify the costs in many institutions.

Thirdly, there are no reported outcomes, which are fundamental in helping surgeons decide as whether to use these implants or not. 

Finally, although lateral compartment UKR is much less common than medial the incidence is increasing and I am not sure that analysis of 30 cases adds anything to the paper to merit inclusion.

On a general overview there are a couple of other minor issues.

 Figure 1 is not clearly defined and adds little to the paper. If it was intended to illustrate the surgical technique then it falls far short id doing so.

On page two , the last sentence at the end of paragraph two is repeated at the start of paragraph three.

I do not like to recommend rejection as I am aware of the amount of work that goes in to the preparation , but unfortunately I do not think this paper merits publication

Author Response

Reviewer 2:

This is essentially a radiographic retrospective review of a large number of medial compartment UKRs and a small number of lateral compartment UKRs using custom made implants. 

It does show less tibial overhang than when using standard implants, which one would expect. It also shows good mid term implant survival using revision as an end point. 

There are a number of issues which diminish the value of this paper.

Firstly the use of CT preoperatively exposes the patient to increased radiation and I have ethical reservations in this regard.

  • We added the following paragraph

One possible concern in PSI is the radiation patients are exposed through the pre-operative computed tomography (CT) scan. Modern CTs have an effective dose between 1 and 7 millisievert (mSv) depending on the organ and the technique. The effective dose for a CT scan of the knee is reported to be 1.3 mSv [31] . In the protocol for PSI a few slides have to be done on the hip joint with a slightly higher dose. The average effective dose through environmental sources is estimated to be 2.4 mSv per year in central Europe rang-ing from 1-10 mSv depending on activity and trhe exact living area. Radiation through medical exposures is on average an extra 2 mSv with variations depending on age and medical condition. The applied radiation of the preoperative CT-scan is not negligible., hHowever it has to be weighed on against the potential advantages of PSI. If the implants lead to lower revision rates there will be a reduced radiation for patients that do not need multiple X-rays before and after revision surgery. Furthermore, in conventional knee ar-throplasty, preop whole leg X-ray is mandatory. The radiation of these images is not neg-ligible tooeither. This radiation is not necessary in patients receiving PSI implants, since the leg axis is also determined through CT. Finally, in the opinion of most experts the risk of developing a disease through CT-scan of the thorax or the abdomen in patients aged over 65 years is negligible [32,33]  So the much lower radiation dose of a CT scan of the ex-tremity is probably irrelevant in these patients. Considering the potential advantages of PSI the necessary radiation for a preoperative CT-scan to plan the PSI implants is justifia-ble in the eyes of the authors.

Secondly , the increased cost must be a consideration in using custom made implants for all cases. I think that any potential benefit would not justify the costs in many institutions.

  • We added the following paragraph

A limitation of the use of PSI is the costs that are 2-2.5-fold higher than for conven-tional implants depending on the country. On the other hand in PSI there is no need for additional trays which reduces costs of sterilization and logistics and saves time. If PSI will lead to reduced revisions, as this is the case in the present study, there is another po-tential of saving money by investing more in the implant during primary surgery. In the future the costs of the implants should also be less by reducing the costs for production through modern 3D printing and an eventually higher use of PSI which should also re-duce costs. Considering all these facts probably the costs are only slightly higher than in conventional implants, although detailed information of the exact extra costs are missing.

Thirdly, there are no reported outcomes, which are fundamental in helping surgeons decide as whether to use these implants or not. 

Finally, the study does not allow conclusions about the functional results since no clinical scores were included in the analysis.

  • We added the following lines in the 2nd paragraph of the discussion

In comparison to these registry data the present study showed favorable results for an individually designed UKA with a revision rate of 4% at 5 years. Furthermore, the most impactful data investigating implant survival are currently retrieved from joint replacement registries, since very large numbers can be assessed over time. However, so far, no registry data have been available for the patient specific UKAs. This emphasizes the importance of performing individual studies with large patient numbers and high follow-up rates. The present study is the largest analysis of the iUni implant with more than 500 cases involved and a follow-up rate of 95.3%. Thus, the present study is -although limited by its retrospective character – the most robust analysis currently available of implant survival of the patient specific UKA .

  • We added the following paragraph in the limitations:

This has not been the aim of the study since implant survival should be investigated. There have been many studies showing excellent clinical results in unicompartmental knee replacement [25,26,34] including the implant used in this study. The patients of this study are followed very closely in the centers after UKA, and .the low revision rates in this study suggests that the clinical results are also satisfactory. Nevertheless, future studies including clinical results are mandatory

Finally, although lateral compartment UKR is much less common than medial the incidence is increasing and I am not sure that analysis of 30 cases adds anything to the paper to merit inclusion.

  • Thank you for this comment which is certainly correct. However, there are not many studies available presenting the results of more than 30 lateral UKAs. To interpret the data correcty, we added the following sentences in the discussion:

The present study confirms the favourable results of the PSI especially in lateral unicompartmental osteoarthritis in a limited number of patients.

 Second, the number of patients receiving a lateral UKA was relatively small. This is due to the significantly rarer indication of lateral UKA. Large case numbers can most likely be obtained with registry analysis, which should also become available for patient specific implants in the future.

 In the conclusion we state the following:

The use of a patient specific implant in this study showed a 100% survival at 4.2 years in 30 lateral knees and these results should be confirmed in the future on a higher number of patients.

On a general overview there are a couple of other minor issues.

 Figure 1 is not clearly defined and adds little to the paper. If it was intended to illustrate the surgical technique then it falls far short id doing so.

  • We added the two sentence to the legend:

Preoperative planning of the prosthesis and surgical guide (Iview) delivered in every patient. It shows the osteophytes that must be removed to position the patient specific instruments (orange colored). The position of the patient specific instruments is also shown. Especially the position of the femoral jig in accordance with the femoral component is very helpful for the surgeon. Furthermore, it shows the final position of the prosthesis (see further details in the text).

On page two , the last sentence at the end of paragraph two is repeated at the start of paragraph three.

  • We changed the sentence to:

The use of a patient specific unicompartmental knee prosthesis should result in a more precise implantation and a better coverage.

I do not like to recommend rejection as I am aware of the amount of work that goes in to the preparation , but unfortunately I do not think this paper merits publication

 We thank you again for your efforts to improve our manuscript and hope we could address all the raised questions sufficiently. If you have any further questions, please don´t hesitate to contact us and we will be happy to provide you with the desired information.

Thank you again for considering our manuscript for your renowned journal,

Yours sincerely,

The authors